# Evaluating the inhibitory efficacy of *Oxalis* phytocompounds on monoamine oxidase B: An integrated approach targeting age related neurodegenerative diseases through molecular docking and dynamics simulations

Ram Lal (Swagat) Shrestha[1,2,3☉], Shiva M. C.[1,2☉], Ashika Tamang[1,2], Manila Poudel[2,4], Nirmal Parajuli[1,2], Aakar Shrestha[2], Timila Shrestha[1,2], Samjhana Bharati[1,2], Binita Maharjan[1,2], Bishnu P. Marasini[2,3,5‡*], Jhashanath Adhikari Subin [ID][2,6‡*]

1 Department of Chemistry, Amrit Campus, Tribhuvan University, Lainchaur, Kathmandu, Nepal,
2 Kathmandu Valley College, Kalanki, Kathmandu, Nepal, 3 Institute of Natural Resources Innovation, Kalimati, Kathmandu, Nepal, 4 Department of Biotechnology, National College, Lainchaur, Kathmandu, Nepal, 5 Nepal Health Research Council, Ministry of Health and Population, Ramshah Path, Kathmandu, Nepal, 6 Bioinformatics and Cheminformatics Division, Scientific Research and Training Nepal P. Ltd., Bhaktapur, Nepal

☉ These authors contributed equally to this work.
‡ These authors also contributed equally to this work.
* subinadhikari2018@gmail.com (JAS); bishnu.marasini@gmail.com (BPM)

## Abstract

Monoamine oxidase B (MAO-B) serves as a critical target in the management of neurodegenerative diseases (NDDs) such as Alzheimer's and Parkinson's due to its role in regulating oxidative stress and dopamine metabolism. In this context, phytochemicals from *Oxalis* species, known for their neuroprotective properties, were explored for their potential MAO-B inhibitory activity using computational approach. Plant-derived compounds, offering a better safety profile than synthetic drugs and greater cost-effectiveness, present a promising avenue for developing alternative therapeutic strategies. Molecular docking (MD), molecular dynamics simulations (MDS), and binding free energy calculations were employed to evaluate the inhibitory potential of *Oxalis* phytochemicals against MAO-B (PDB ID: 4A79). Stable ligand-protein complexes with optimal docking scores were selected, and key parameters from molecular dynamics trajectories, including binding stability and interactions, were analyzed to identify high potential inhibitors of MAO-B for therapeutic development. Results showed beta-sitosterol (−11.92 kcal/mol), squalene (−11.89 kcal/mol), etretinate (−11.46 kcal/mol), rhoifolin (−11.44 kcal/mol), and swertisin (−11.13 kcal/mol) demonstrated superior binding affinities compared to the native ligand (−11.12 kcal/mol). Three additional compounds; phloridzin (−11.10 kcal/mol), rhapontin (−11.02 kcal/mol), and diosmetin 7-O-beta-D-glucopyranoside (−10.96 kcal/mol) exhibited better binding than reference drugs. The predominant interactions between

**Data availability statement:** All relevant data are within the manuscript and its Supporting Information files.

**Funding:** The author(s) received no specific funding for this work.

**Competing interests:** The authors have declared that no competing interests exist.

protein and ligand were hydrophobic, with hydrogen bonds and Pi-stacking enhancing the complexes' stability. The evaluation based on geometrical and thermodynamic metrics derived from 200 ns MDS, identified rhoifolin, beta-sitosterol, and swertisin as promising MAO-B inhibitors. Minimal translational and rotational movements of these ligands within the catalytic site of MAO-B under quasi-physiological conditions suggested effective inhibition. Preserved thermodynamic feasibility reinforced these findings. ADMET analysis identified squalene and beta-sitosterol as CNS active candidates with favorable pharmacokinetics, while etretinate, rhoifolin, and swertisin may act as peripheral modulators, requiring optimization for improved CNS delivery. Further experimental validation of efficacy, pharmacokinetics, and safety is recommended to advance the therapeutic potential of these hit candidates.

## 1. Introduction

Neurodegenerative diseases (NDDs) are neurological disorders characterized by the progressive loss and dysfunction of neurons, synapses, and glial cells within the Central Nervous System (CNS) or Peripheral Nervous System (PNS), leading to cognitive, motor, and sensory impairments [1]. Affecting approximately 15% of the global population [2], with aging as a primary risk factor, NDDs are projected to become the second most prevalent cause of mortality in the coming decades [3–5]. Common disorders include Alzheimer's, Parkinson's, Huntington's, multiple sclerosis, and amyotrophic lateral sclerosis (ALS), with Alzheimer's and Parkinson's being the most common ones [5]. NDDs are marked by pathological protein aggregations such as Amyloid-beta (Aβ), hyperphosphorylated Tau, α-Synuclein, Huntingtin, TAR DNA-binding protein 43 (TDP-43), Prion protein (PrP) etc. [6]. Despite varying etiologies, these conditions exhibit selective brain vulnerability, and shared pathological mechanisms with clinical manifestations such as abnormal proteostasis, oxidative stress, neuroinflammation, and neuronal loss, all of which play significant roles in the progression of these disorders [1]. A hallmark of NDDs is the dysfunction of monoaminergic pathways, particularly the upregulation of monoamine oxidase B (MAO-B), which has been linked to neuronal degeneration [1,7–10]. Elevated MAO-B activity increases oxidative damage and neuronal death, exacerbating the neurodegenerative process [10]. These mechanisms highlight the need for targeted therapies to address both protein aggregation and monoaminergic dysregulation in these diseases.

MAO-B is a flavoenzyme located in the outer membrane of the mitochondria [11,12]. It is one of two isozymes of monoamine oxidase, the other being Monoamine Oxidase A (MAO-A) [13]. Both isozymes are involved in the oxidative deamination of monoamines [14], but they differ in substrate specificity, tissue distribution, and their roles in neurotransmitter metabolism [12,13,15,16]. MAO-B is primarily found in platelets, astrocytes, and hepatocytes [14] preferentially deaminating substrates such as 2-phenylethylamine and benzylamine [15] and can also deaminate other substrates

including, dopamine, noradrenaline, adrenaline, tryptamine, and tyramine [17,18]. Fig 1 shows the crystal structure of MAO-B enzyme in its holo form, represented as a ribbon representation.

MAO-B enzyme levels are found to naturally increase with age due to heightened neuronal cell death, disrupting monoaminergic signaling by depleting the neurotransmitters and generating neurotoxic byproducts like hydrogen peroxide [14]. This byproduct triggers the formation of reactive oxygen species (ROS), leading to oxidative stress, mitochondrial damage, and neuronal apoptosis, further exacerbating neuronal cell death and contributes to both protein misfolding and aggregation [10]. In Alzheimer's disease (AD), MAO-B expression is notably upregulated in the hippocampus and cerebral cortex, particularly in reactive astrocytes surrounding amyloid-β plaques [12,15]. MAO-B inhibitors, such as selegiline and rasagiline, have been clinically validated for their efficacy in slowing disease progression and improving symptoms in neurodegenerative conditions, particularly Parkinson's disease [19], by stabilizing dopamine levels, providing symptomatic relief, and offering potential neuroprotective effects [20]. Thus, MAO-B is a promising therapeutic target for managing neurodegenerative disorders by restoring monoaminergic balance through its inhibition [21,22].

Natural phytochemicals are gaining prominence as viable alternatives to synthetic drugs due to their unique pharmacological properties, structural diversity, and historical validation through traditional medicine [23]. Approximately 40% of their chemical scaffolds are absent in synthetic libraries, and 45% of today's bestselling drugs originate from natural products or their derivatives [23]. Despite advancements in synthetic chemistry, natural products remain a crucial source of novel structures, offering lower toxicity, cost-effectiveness, and multifaceted therapeutic actions, making them essential in the search for safer and more effective treatments for complex diseases [24,25].

*Oxalis*, a perennial herb from the Oxalidaceae family, is widely distributed in tropical and subtropical regions [26]. Known for its diverse phytochemical profile, including flavonoids and phenolics, *Oxalis* has demonstrated antioxidant, anti-inflammatory, and neuroprotective properties [27], making its bioactive compounds promising candidates for age-related NDDs [28,29]. These phytochemicals have been explored for their ability to mitigate oxidative stress, inflammation, and neuronal damage, the key factors in NDDs progression [30–34]. Notably, several studies have reported MAO-B inhibitory activity in natural products structurally similar to those found in *Oxalis*, including flavonoids and polyphenols derived from other plants [35,36]. However, few studies have examined the mechanistic, geometrical, and thermodynamic aspects of *Oxalis* phytochemicals, especially against MAO-B, a critical enzyme involved in NDDs. Understanding the molecular interactions between bioactive compounds and biological targets through these analyses is vital for gaining insights into ligand-protein binding stability, thermodynamic favorability, and the driving forces behind these interactions.

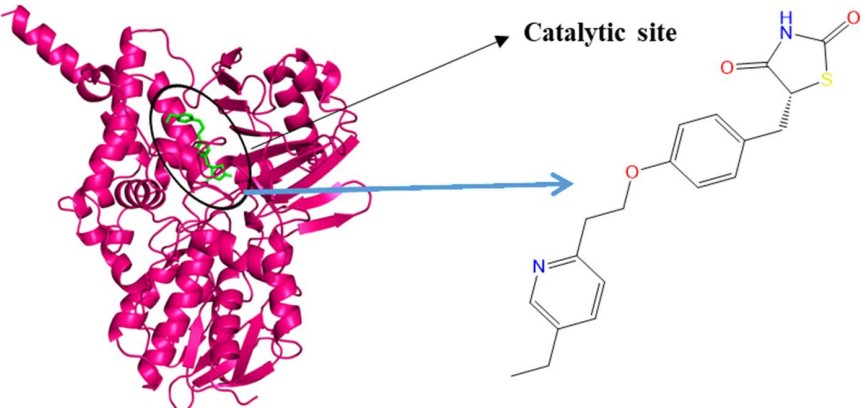

**Fig 1. Monoamine Oxidase-B (MAO-B) enzyme structure in its holo form depicted in ribbon representation, with native ligand P1B, Pioglitazone, shown as bond line model at the orthosteric site.**

This knowledge not only deepens our understanding of enzyme modulation but also supports the rational design of novel, targeted therapeutic agents. Computational methodologies, including virtual screening, molecular docking, and dynamics simulations, are integral to drug discovery, particularly during the preliminary stages of research. These *in silico* approaches expedite the identification and optimization of potential drug candidates by predicting their interactions with specific biological targets prior to laboratory synthesis and subsequent biological evaluation. They serve as preliminary, faster, more cost-effective, and high-throughput filters compared to traditional bench work, which involves hands-on experimental techniques such as compound synthesis, biochemical and cellular assays, and animal studies that directly measure biological activity, toxicity, and pharmacokinetics. By guiding and refining experimental design, computational methods assist with efficient experimental design creating a more discerning and focused pool of possible therapeutic strategies. These predicted targets can then be biologically validated through *in vitro* assays such as enzyme inhibition or cell-based functional studies followed by *in vivo* experiments to confirm their therapeutic efficacy and safety.

Molecular Docking (MD) predicts ligand binding modes within protein structures, supporting virtual screening and generating hypotheses for target inhibition, crucial for lead optimization in drug discovery [37]. Molecular dynamics simulations (MDS) model the movement of atoms in molecular systems, capturing processes such as ligand binding, protein folding, and conformational changes at femtosecond resolution. MDS can also predict atomic-level responses to alterations like mutations or phosphorylation [38]. Combined, these tools provide a detailed understanding of ligand-protein interactions and biomolecular dynamics, facilitating the rational design, optimization, and experimental validation of drug candidates.

Despite extensive research, the development and underlying processes of NDDs remain poorly understood, with no effective cure currently available. The present study aims to identify potential inhibitors of the MAO-B enzyme, a key target in age-related NDDs, utilizing computational tools such as MD and MDS.

## 2. Computational method

### 2.1. Preparation of ligand database

A database of 36 phytochemicals, all with molecular weights in the range of 500 g/mol was created based on previously reported compounds from two *Oxalis* species; *Oxalis corniculata* and *Oxalis latifolia*. [39–44]. This molecular weight range aligns with established pharmacological trends indicating that smaller molecules are more likely to fit into the target protein's binding site with minimal steric hindrance [45–47] and are considered suitably small for effective interaction. These are often associated with favorable pharmacokinetics, thereby increasing the likelihood of later clinical success [48].

The chemical structures were obtained from the PubChem database (https://pubchem.ncbi.nlm.nih.gov/) [49] in SDF format. The structures were converted into PDB format using the Avogadro program version 1.20 [50] and the molecular formulas were validated introducing necessary hydrogen atoms. Energy minimization of the molecular structures was carried out using the Universal Force Field (UFF), with a maximum iteration limit of 5000 steps employing the conjugate gradient optimization algorithm. An energy convergence threshold of $1.0 \times 10^{-8}$ units was used, and the minimization procedure was repeated until no further significant alterations in atomic positions were detected. The bond order, with particular emphasis on stoichiometry verification, was systematically assessed for each molecule, confirming the absence of steric hindrance.

### 2.2. Target selection

Integrating experimental data through comprehensive literature analysis facilitates the efficient identification of phytochemical targets. In this study, MAO-B was selected as the target enzyme due to its pathophysiological role in catalyzing the oxidative deamination of neurotransmitters, which contributes to elevated oxidative stress and neuronal degeneration, key factors in the progression of NDDs.

## 2.3. Preparation of target geometry

The 3D crystal structure of MAO-B enzyme with PDB ID of 4A79 (https://doi.org/10.2210/pdb4A79/pdb) (X-Ray resolution = 1.89 Å, Classification = oxidoreductase, Expression System = *Komagataella pastoris*, Organism = *Homo sapiens*) was selected. Only the chain A, with the native ligand, pioglitazone was considered and the other chain B was deleted. The native ligand, co-ligands, water molecules and ions were removed, and polar hydrogen atoms were introduced to the molecular structures. The resulting structure was exported in apo form in PDB format using the PyMOL software [51] for molecular docking calculation.

## 2.4. Molecular docking calculations

Molecular docking is essential in structure-based drug discovery but is limited by factors such as rigid receptor-ligand treatment, inadequate modeling of entropy and solvation, and the limited accuracy of scoring functions [52]. To address this inherent limitation of molecular docking, the DockThor portal (https://dockthor.lncc.br/v2/) [53] was employed to predict the optimal ligand binding pose at the receptor's orthosteric site leveraging its flexible docking capabilities. DockThor implements a soft docking protocol that incorporates both protein and ligand flexibility within a hydrated environment, enabling a more accurate representation of induced fit and dynamic molecular interactions for determining the optimal binding pose of the ligand at the orthosteric site of the receptor [54].

Docked conformations were ranked based on total interaction energy, and predicted binding affinities were evaluated using the DockTScore scoring function. The docking score, expressed in kcal/mol, provides an approximation of ligand-receptor binding affinity by reflecting the overall strength of their predicted interactions [52]. It comprises contributions from van der Waals interactions, electrostatic interactions, hydrogen bonding (implicitly), solvation effects (approximate), and torsional energy associated with ligand conformational flexibility. While limitations associated with the scoring function persist, as is common with most docking tools, DockThor offers a suitable compromise between biological relevance and computational efficiency. Its open-access framework, flexible docking methodology, and high-throughput capabilities made it a fitting choice for the present study [54].

For the orthosteric site, a grid center at coordinates (51, 157, 31), a grid dimension of (15, 15, 15) Å$^3$, a discretization of 0.16, and a total of 830,584 grid points were utilized. The genetic algorithm parameters were configured with 1,000,000 evaluations of the scoring functions, a population size of 750, and 24 runs. Structural analysis and visualization for the best scoring candidates were carried out by using the PLIP server and PyMOL software [55].

## 2.5. Molecular dynamics simulation (MDS)

Molecular dynamics simulations were performed for the most promising receptor-ligand complexes, which exhibited the most favorable binding scores, determined by molecular docking studies. These simulations were conducted using the GROMACS software version 2021.2 [56]. The CHARMM27 force field [57] was applied to model the protein, while the ligand parameters were derived from the SwissParam server (http://swissparam.ch/) [58]. A triclinic simulation box with a spacing of 12 Å was employed for the adducts, in a solvent environment modeled with TIP3P water. An isotonic NaCl solution (0.15 M) was introduced, and the system was neutralized by adding the appropriate quantities of counter ions (either Na$^+$ or Cl$^-$) to ensure that the ionic environment accurately reflects physiological conditions. The temperature was set at 310 K, and the pressure was maintained at 1 bar. The temperature regulation and the pressure coupling was done by V-rescale weak coupling method and Parrinello-Rahman method respectively. Equilibration was performed with, two NVT and two NPT ensembles in 4 stages, each for a duration of 1 ns. A time step of 2 fs was selected, with simulation time of 200 ns without any constraints, in accordance to the procedure specified in the literature [59]. The built-in modules of the GROMACS program were utilized to extract various geometrical parameters such as Root Mean Square Deviation (RMSD), Root Mean Square Fluctuation (RMSF), Solvent Accessible Surface Area (SASA), Radial Pair Distribution Function (RPDF), Radius of Gyration (R$_g$), Hydrogen bond count, distribution, and modulation from the MDS trajectory.

### 2.6. Spontaneity of receptor-ligand adduct formation

The change in binding free energy at the final configuration of the system was evaluated by analyzing the equilibrated portion of the MDS trajectory, which comprised of 200 frames over a duration of 20 nanoseconds. The estimation was performed using the MMPBSA module [60], which utilized the Poisson-Boltzmann solvation model, as implemented within the GROMACS software suite. The change in binding free energy is delineated by the following equation:

$$\Delta G_{BFE} = G_{RL} - G_R - G_L \tag{1}$$

where, $\Delta G_{BFE}$ = Change in binding free energy of the adduct formation, $G_{RL}$ = binding free energy of receptor ligand complex, $G_R$ = binding free energy of receptor, $G_L$ = binding free energy of ligand

The gmx_MMPBSA_ana subprogram was utilized for generating the plots and for conducting the data analysis.

### 2.7. ADMET prediction

To assess the drug-like properties, safety, and pharmacokinetics of the top-scoring ligands from molecular docking and molecular dynamics studies, the pkCSM (https://biosig.lab.uq.edu.au/pkcsm/) server [61] was employed. This tool predicted essential ADMET parameters including absorption, distribution, metabolism, excretion, and toxicity. Integrating these predictions with docking and molecular dynamics results enabled a comprehensive evaluation of the most promising ligands, supporting their potential as safe and effective drug candidates for further development.

### 2.8. Computational resources

All computations were conducted with multiprocessor systems (24 cores) with GPU support (24 GB) running on Ubuntu 20.04 LTS. Data analysis and visualization were performed on a PC operating on Windows 10.

## 3. Results and discussion

### 3.1. Molecular docking protocol validation

The root mean square deviation (RMSD) between the crystal structure and the predicted structures are commonly employed to verify whether the docking simulation has accurately predicted a close-matching pose for the docked ligand [62]. The docking protocol was validated by ensuring that the RMSD of the heavy atoms between the docked ligand, obtained from the calculations, and the native ligand in the crystal structure was below 2 Å as depicted in the Fig 2.

### 3.2. Docking scores and comparison

The binding affinities (docking scores) of top eight phytochemicals, along with the native ligand and reference drugs in their most favourable binding conformations, against the target enzyme, MAO-B are ranked in ascending order in Table 1 (the more negative the docking score, the stronger the binding).

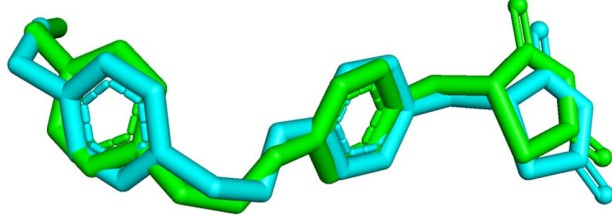

**Fig 2. Superimposition of docked ligand (green) derived from the molecular docking calculations, with the native ligand (cyan) present in the crystal structure (heavy atom RMSD = 0.768 Å).**

**Table 1. Binding affinities of top 8 compounds along with native ligand and reference drugs considered.**

| Compounds | PubChem CID | Binding affinities (kcal/mol) |
|---|---|---|
| **Beta-sitosterol** | **222284** | **−11.92** |
| Squalene | 638072 | −11.89 |
| Etretinate | 5282375 | −11.46 |
| Rhoifolin | 5282150 | −11.44 |
| Swertisin | 124034 | −11.13 |
| Phloridzin | 6072 | −11.10 |
| Rhapontin | 637213 | −11.02 |
| Diosmetin 7-O-beta-D-glucopyranoside | 11016019 | −10.96 |
| Pioglitazone (Native ligand) | 4829 | −11.12 |
| Safinamide (reference drug) | 131682 | −9.86 |
| L-deprenyl (reference drug) | 26757 | −8.80 |
| Rasagiline (reference drug) | 3052776 | −8.37 |

Notably, 5 phytocompounds, beta-sitosterol, squalene, etretinate, rhoifolin, and swertisin demonstrated superior binding affinities (−11.13 to −11.92 kcal/mol) exceeding the binding affinity of the native ligand (−11.12 kcal/mol) and those of the reference drugs considered (−8.37 to −9.86 kcal/mol). The hit candidates could be considered to be strongly bound to the target protein than drugs and could perform better. The inhibition could possibly be more effective. The three phytocompounds, phloridzin (−11.10 kcal/mol), rhapontin (−11.02 kcal/mol), and diosmetin 7-O-beta-D-glucopyranoside (−10.96 kcal/mol), exhibited binding affinities comparable to or slightly lower than that of the native ligand, yet superior to those of the reference drugs. Despite this, their similar interaction profiles with other potent ligands suggests, they may still exhibit significant inhibitory potential against MAO-B. This is further explored in the subsequent section, where they are highlighted as additional promising candidates for further exploration in therapeutic applications targeting NDDs.

The chemical structures of the top-ranked phytochemicals identified through molecular docking analyses are presented in Fig 3.

### 3.3. Molecular interactions in different adducts

Protein-ligand interactions are fundamental to biological processes such as cell transduction, receptor modulation, enzyme action, and biochemical pathways, and even in drug discovery process [63,64]. A detailed breakdown of these interactions is crucial for designing effective therapies and drugs [63]. By analyzing binding sites and interaction patterns using Interaction Visualization Tools, profundities into the modulation of biological systems for therapeutic purposes can be achieved. This foundational step enables the identification of key interactions vital for enzyme functionality, providing a basis for the design of targeted strategies to modulate enzyme activity and inform the development of potential therapeutic interventions. S1 Fig in S1 File. (in the supplementary information) illustrates the amino acid residues involved in ligand interactions across various complexes, accompanied by a detailed list of their interaction specifics in Table 2.

Hydrophobic interactions involving key amino acid residues **TYR60, PHE103, PRO104, TRP119, LEU171, ILE198, ILE199, GLN206, ILE316, TYR326,** and **PHE343** were consistently observed across all top-ranked ligand–MAO-B complexes, including the native ligand-bound structure. These same residues were identified as components of the MAO-B binding site based on structural data from PDB ID: 4A79, as annotated by the BioLiP server (https://zhanggroup.org/Bio-LiP/). Moreover, the literature [65–68] also reported these residues as part of the enzyme's active site cavity. Collectively, these findings underscore the crucial role of non-polar interactions in maintaining the structural integrity of the MAO-B binding pocket and mediating effective ligand recognition and stabilization [65]. This reinforces the reliability of the docking results and highlights the pivotal role of these residues in stabilizing binding site interactions, with potential implications for

**Fig 3. Chemical structures of top 8 phytochemicals from *Oxalis* species identified based on molecular docking scores against MAO-B enzyme.**

enzyme activity and substrate specificity. Notably, hydrophobic interactions were found as the predominant type of interactions in Complex 1, Complex 2, Complex 3, and Complex 5 (as illustrated in the protein-ligand interaction profiling presented in S1 Fig in S1 File. of the Supplementary information). In addition to the aforementioned residues, specific amino acids contributed significantly to hydrophobic interactions in individual complexes like LEU164, LEU167, PHE168, and TYR398 in Complex 1, LEU88, LEU164, LEU167, PHE168, TYR398, and TYR435 in Complex 2, LEU164 in Complex 3, and TYR398 in Complex 5. In Complex 4, a strong hydrogen bond was observed with PRO102 (2.41 Å), while moderate

**Table 2. Overview of non-covalent interactions and corresponding distances between the docked ligand and key amino acid residues in different protein-ligand complexes.**

| Ligands | Type of Interactions | Orthosteric site amino acid residues (Distance, Å) |
|---|---|---|
| Beta-sitosterol | Hydrophobic | **PHE103** (3.26), **PRO104** (3.34), **TRP119** (3.70, 3.62), LEU164 (3.20), LEU167 (3.32), PHE168 (3.47), **LEU171** (2.73, 3.10), **ILE 198** (3.99), **ILE199** (3.27, 3.10), **GLN206** (3.14), **TYR326** (3.36), TYR398 (3.20) |
| Squalene | Hydrophobic | **TYR60** (3.61), LEU88 (3.58), **PRO104** (2.89), **TRP199** (3.65, 2.98), LEU164 (2.95, 3.10), LEU167 (2.87), PHE168 (3.41, 3.17), **LEU171** (3.26, 3.05), **ILE199** (3.27, 2.83, 3.88), **GLN206** (3.82), **ILE316** (3.43), **TYR326** (3.47), **PHE343** (3.39), TYR398 (3.82), TYR435 (2.90, 3.70) |
| Etretinate | Hydrophobic | **PHE103** (3.28), **PRO104** (3.17), **TRP119** (3.57, 3.47), LEU164 (2.84, 2.95), **LEU171** (3.21, 3.91), **ILE199** (3.66, 3.14, 3.33), **ILE316** (3.89), **TYR326** (3.94) |
| Rhoifolin | Hydrophobic | **TYR60** (3.54), **PHE103** (3.41), **ILE199** (3.85, 3.48, 3.92), **GLN206** (3.55), **PHE343** (3.76), TYR435 (3.83) |
| | Hydrogen bond | GLU84 (2.65), PRO102 (2.41), LEU164 (3.80), THR201 (3.67) |
| | Pi-stacking | **TYR326** (4.58) |
| Swertisin | Hydrophobic | **LEU171** (3.53), **ILE199** (3.45), **GLN206** (3.63), **TYR326** (3.82, 3.46), **PHE343** (3.73), TYR398 (3.79) |
| Phloridzin | Hydrophobic | **TRP119** (3.93), LEU164 (3.48, 3.89, 3.16), LEU167 (3.57), PHE168 (3.28, 3.35), **ILE199** (3.39), **ILE316** (3.51), **TYR326** (3.49) |
| | Hydrogen bonds | TYR435 (3.70) |
| | Pi-stacking | **TRP119** (3.91) |
| Rhapontin | Hydrophobic | **TYR60** (3.54), **LEU171** (3.31), **ILE199** (3.97), **GLN206** (3.58), TYR398 (3.88) |
| | Hydrogen bonds | PRO102 (2.43, 3.12), PHE168 (2.64), CYS172 (3.85), **ILE199** (3.01) |
| | Pi-stacking | **TYR326** (5.10) |
| Diosmetin 7-O-beta-D-glucopyranoside | Hydrophobic | LEU88 (2.95), PHE99 (3.87), **ILE199** (3.64), **ILE316** (3.56) |
| | Hydrogen bonds | GLU84 (2.65), THR201 (2.98), TYR435 (3.02, 2.76) |
| Native ligand, Pioglitazone | Hydrophobic | **PHE103** (3.78, 3.46), **TRP119** (3.58, 3.49, 4.00), **LEU171** (3.17), **ILE199** (3.48, 3.39, 3.77), **ILE316** (3.60) |
| | Hydrogen bonds | TYR435 (3.16) |
| Rasagiline (Reference drug) | Hydrophobic | LEU88 (3.50), PRO102 (3.95), PHE168 (3.59), **LEU171** (3.33), **ILE199** (3.73, 3.49), **ILE316** (3.52) |
| | Hydrogen bonds | **ILE199** (2.92) |
| Safinamide (Reference drug) | Hydrophobic | **TRP199** (3.95), PHE168 (3.67), **LEU171** (3.28), **ILE199** (3.94, 3.67, 3.63), **GLN206** (3.90), **PHE343** (3.67) |
| | Hydrogen bonds | PRO102 (3.10) |
| L-Deprenyl (Reference drug) | Hydrophobic | PHE168 (3.73), **LEU171** (3.30, 3.53), **ILE199** (3.81, 3.58), **TYR326** (3.27), TYR398 (3.87), TYR435 (3.67) |

hydrogen bonds were found with GLU84 (2.65 Å), LEU164 (3.80 Å), and THR201 (3.67 Å) in ligand binding interactions. TYR326 at a distance of 4.58 Å was found to be involved in Pi-stacking. TYR435 was identified as additional amino acid residue involved in hydrophobic interaction in complex 4. In Complex 6, a weak hydrogen bond with TYR435 at a distance of 3.70 Å was identified. TRP119 at a distance of 3.91 Å was found to be involved in Pi-stacking. Hydrophobic interactions with LEU164, LEU167, and PHE168 were also significant. In Complex 7, PRO102, PHE168, CYS172, ILE199, were found to be involved in strong to moderate hydrogen bonding. Pi-stacking was observed with TYR326 (5.10 Å), and TYR398 contributed to hydrophobic interaction. In Complex 8, the ligand formed hydrogen bond with GLU84, THR201, and TYR435, while LEU88 and PHE99 were notable contributors to hydrophobic interactions. In the docked native ligand complex, TYR435 formed a moderately strong hydrogen bond at a distance of 3.16 Å, with most of the previously identified residues contributing to hydrophobic interactions. The reference MAO-B inhibitors; rasagiline, safinamide, and L-deprenyl

exhibited similar binding patterns, predominantly interacting with residues ILE199 and LEU171 through hydrophobic contacts. In addition to these common interactions, rasagiline formed a hydrogen bond with ILE199, while safinamide established a hydrogen bond with PRO102. L-deprenyl, in contrast, was found to involve only in hydrophobic interactions, without forming any hydrogen bonds.

The consistent involvement of key amino acid residues within the orthosteric site demonstrated by shared hydrophobic interactions observed with the native ligand and reference MAO-B inhibitors (rasagiline, L-deprenyl, and safinamide) highlights their fundamental role in ligand recognition and binding. Importantly, the top phytochemical candidates not only involved these conserved residues but also formed additional hydrophobic contacts and hydrogen bonds with other surrounding amino acids. The presence of other interactions, including Pi-stacking, may enhance the overall binding affinity and stability of these ligands within the active site. This expanded interaction network may enhance the candidate phytochemicals' capacity to effectively disrupt or inhibit MAO-B's normal biological function by strengthening binding site occupancy [69].

While hydrophobic interactions may play a key role in initial ligand binding, the inclusion of other types of interactions is likely to contribute to enhanced overall binding stability, specificity, and functional dynamics, making them essential considerations in molecular docking and drug design strategies [70,71]. Although the complexes exhibited promising binding affinities and interactions based on the static docking results, their strength and stability must be assessed over time. This evaluation has been conducted using molecular dynamics simulations, the findings of which are discussed in the subsequent section.

### 3.4. Geometrical analysis of adducts conformation

The persistence of structural and geometrical stability of protein-ligand adducts can be inspected through multiple metrics extracted from MDS trajectories. The adduct is considered more stable if the ligand stays bound to the orthosteric site for a longer duration while maintaining its orientation and location relative to the protein backbone [72,73].

A spatial examination of the ligand at the receptor's orthosteric site was conducted using snapshots taken at various time points to assess its translational and rotational movements throughout the production phase. The stability and sustained binding of the ligand were evaluated to assess its potential as an effective inhibitor. Fig 4 shows the snapshots captured at (1, 50, 100, 150, 200) ns of whole simulation period, illustrating the positions of the ligands at the orthosteric site of the protein and the conformation of protein geometry around the orthosteric site in top five protein-ligand adducts.

In first complex, beta-sitosterol exhibited only minor orientation changes at 50, 100, 150, and 200 ns compared to its position at 1 ns in the adduct. The ligand demonstrated the minimal translational movement, as evidenced by the consistently low and stable RMSD profile that is to be explored in more detail in the RMSD discussion section. While the core structures of both the protein and ligand remained largely unchanged, the inherent flexibility of the loops near the ligand facilitated their movement to avoid steric clashes and optimize interactions. This flexibility allowed the ligand to undergo slight orientation shifts within the binding pocket, most notably at 50 ns, as observed in the corresponding snapshot. In the second complex, minor loop repositioning and subtle adjustments in nearby alpha-helices were observed at 50 ns, accompanied by slight torsional changes in squalene to enhance binding. The same location of squalene at 100, 150, and 200 ns is further corroborated by the plateau observed in its RMSD curve, indicating stable binding. In the third complex, etretinate, remained at nearly the same location throughout the simulation, maintaining a consistent orientation at 50, 100, 150, and 200 ns, with a slight deviation observed at 1 ns, indicating an initial settling phase. The protein backbone maintained stable orientations until around 150 ns, after which movement in the alpha helices was observed, that led to minor adjustments to the protein structure. In the fourth complex, rhoifolin exhibited a minor change in its orientation at 50 ns, which was maintained until 100 ns. A further slight shift in orientation occurred at 150 ns, which remained stable until the end of simulation at 200 ns. The protein maintained a conserved position throughout, indicating stable ligand-protein interaction with minimal conformational changes towards the end of the simulation. In the fifth complex, both the protein

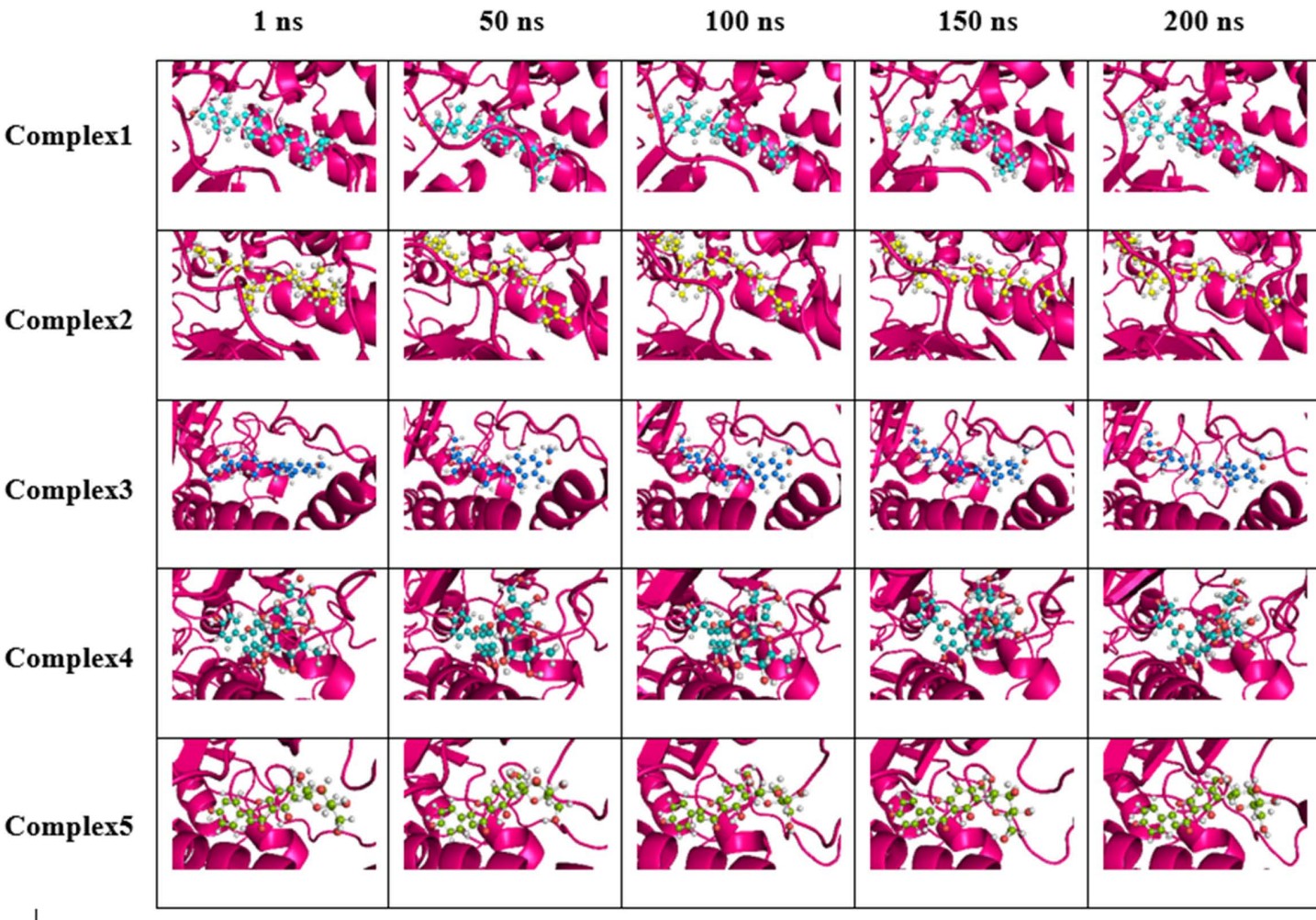

**Fig 4. Visualization of the top five protein-ligand adducts at various time points during molecular dynamics simulations, highlighting the dynamical nature of the ligand relative to the protein structure and also of the protein geometry around the active site.**

and the ligand, swertisin, maintained a consistent orientation throughout the simulation from 1 ns to 200 ns. This stability indicated a persistent binding interaction throughout this time frame, as evidenced by the consistently smooth RMSD curve (Fig 5).

The snapshots confirmed stable ligand binding across all complexes, with minimal conformational changes. This indicates the conservation of orientation and location of the ligand in the complexes. Ligand-protein interactions remained consistent, and the protein structure displayed some flexibility in key regions to optimize binding, as further discussed in the subsequent sections. Quantitative details, including mathematical parameters, are presented to evaluate complex stability, complementing the visual insights from the snapshots taken at different time points of the MDS trajectory.

**3.4.1. Root mean square deviation (RMSD).** RMSD is a crucial parameter for evaluating the stability of protein-ligand complexes. A higher RMSD suggests greater deviation from the reference conformation over time, while a stable RMSD suggests system equilibration, and persistent fluctuations may imply ongoing structural adjustments [74,75]. The

stability of both the top 8 ligands (ligand RMSD) and the protein backbone (backbone RMSD) was carefully analyzed during the simulation, which is presented in Fig 5 and in supplementary information (S2 Fig in S1 File.), respectively.

Beta-sitosterol (violet) showed an initial rise around 8 ns as the ligand adjusted to the binding site, followed by a plateau until 175 ns with a negligible fluctuation at 175 ns, indicated nearing stabilization, supported by a low average RMSD of (0.26 nm ± 0.04 nm). This suggested minimal ligand movement and a stable binding pose. Squalene (green) exhibited an initial rise in RMSD within the first 10 ns, followed by a stable plateau, indicating consistent stability. Etretinate (red) showed a gradual rise with minor fluctuations toward the end, suggesting slight binding adjustments. The average RMSD were 0.66 ± 0.04 nm for squalene and 0.66 ± 0.05 nm for etretinate. Rhoifolin (cyan) exhibited an initial fluctuation in its RMSD curve, followed by a smooth decline until 145 ns, after which the curve elevated steadily, maintaining a smooth trajectory until the end of the simulation. The average RMSD for rhoifolin was 0.24 ± 0.06 nm. The upward shift in the RMSD after 145 ns could be attributed to the rotational motion of the ligand while swertisin (magenta) exhibited a smooth, stable curve with an average RMSD of 0.27 ± 0.03 nm with minimal fluctuation with no any orientation change of the ligand, as discussed above (Fig 4).

For the ligand-complexes with binding affinities (docking scores) slightly lower than the native ligand, phloridzin (blue) exhibited a moderate RMSD curve with minimal fluctuations up to 105 ns, yielding an average RMSD of 0.40 ± 0.03 nm. In contrast, rhapontin (orange) and diosmetin 7-O-beta-D-glucopyranoside (maroon) displayed nearly flat RMSD curves with minimal fluctuations, with average RMSD of 0.26 ± 0.03 nm and 0.25 ± 0.05 nm, respectively.

In conclusion, based on the RMSD analysis, rhoifolin, beta-sitosterol, and swertisin demonstrated the most stable binding with low RMSD and minimal fluctuations, making them the strongest candidates for further study and potential inhibitors of receptor function. Squalene and etretinate exhibited slight flexibility but maintained stable binding, suggesting effective binding despite minor conformational adjustments. Rhapontin and diosmetin 7-O-beta-D-glucopyranoside showed stable binding, though with slightly higher RMSD than the top candidates. In contrast, phloridzin exhibited weaker binding with higher RMSD, indicating less stability in its complex. However, all ligands exhibited stable binding at the active site while maintaining their conserved poses, suggesting their potential to modulate the target protein.

**3.4.2. RMSD of protein backbone.** The RMSD of apo protein (black) with respect to the protein backbone, showed a relatively stable trajectory with minimal fluctuations upto *ca.* 100 ns of the simulation followed by the steady curve during the final 100 ns with an average RMSD of 0.31 ± 0.14 nm. In the bound state, the protein exhibited varying RMSD depending on the ligand. All the protein geometries complexed with ligands exhibited moderately smooth curves, with minor fluctuations observed in each individual protein. Slightly higher fluctuations were noted in the proteins complexed with etretinate and rhoifolin. This can be evidenced from the snapshots discussed earlier. The protein complexed with phloridzin (0.21 ± 0.03 nm) demonstrated significantly lower RMSD than the apo protein, indicating enhanced stability and

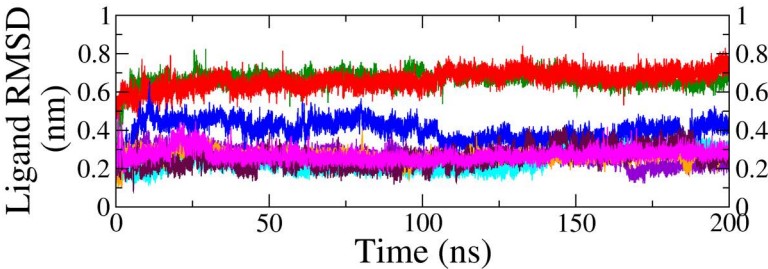

**Fig 5. RMSD of top 8 ligands relative to protein backbone obtained from MDS trajectories of different complexes (violet = beta-sitosterol, green = squalene, red = etretinate, cyan = rhoifolin, magenta = swertisin, blue = phloridzin, orange = rhapontin, maroon = diosmetin-7-O-beta-D-glucopyranoside).**

minimal structural deviations upon ligand binding. The protein complexed with beta-sitosterol (0.26±0.04 nm), diosmetin-7-O-beta-D-glucopyranoside (0.23±0.03 nm), and rhapontin (0.23±0.05 nm) also showed lower RMSD than the apo protein, suggesting that these ligands stabilize the protein structure effectively. Protein with squalene (0.28±0.05 nm) and swertisin (0.28±0.05 nm) exhibited RMSD close to that of the apo protein, indicating comparable stability with slight structural adjustments upon binding. The protein complexed with etretinate (0.32±0.05 nm) and rhoifolin (0.37±0.08 nm) exhibited slightly higher RMSD compared to the apo protein, although the difference was not remarkable. The RMSD curve of protein backbone complexed with top 8 ligands is presented in S2 Fig in S1 File. (in the supplementary information).

The RMSD analysis of protein backbone demonstrates that it remained stable across all ligand complexes, with no significant geometric changes or structural deviations. This indicates that the protein maintained its overall stability, with its conformation largely unaffected by ligand binding. Consequently, the target protein is considered druggable, as it can effectively bind ligands at its catalytic site, leading to the formation of a stable adduct.

### 3.4.3. Root mean square fluctuation (RMSF).

RMSF quantifies the flexibility of individual atoms or residues in a molecular system by evaluating their variations from average positions over time in molecular dynamics simulations [53]. This analysis provides insights into the dynamic behavior of the protein during the simulation, highlighting regions of flexibility and rigidity within the structure. Reduced fluctuations in the positions of alpha-carbon atoms suggest that the protein backbone maintains stability, which correlates with higher structural integrity within the complex [76]. S3 Fig in S1 File. (in the supplementary information) illustrates the RMSF analysis of the alpha-carbon atoms of the amino acid residues in the top 8 protein-ligand complexes, along with that of the apo protein.

All the protein backbone of top eight complexes exhibited RMSF below *ca.* 0.3 nm with similar curve pattern to that of apo form (below *ca.* 0.3 nm) with no significant differences. This overall trend suggested that alpha-carbon fluctuation has no significant effect on ligand binding, indicating the structural integrity of the protein-ligand system.

### 3.4.4. Solvent accessible surface area (SASA).

The solvent accessible surface area (SASA) represents the surface area of a macromolecule that is accessible to solvent molecules. It is a critical parameter in understanding molecular interactions, folding, and stability in protein-ligand complexes [77,78]. The exposure of hydrophobic regions to water is energetically unfavorable, and minimizing this exposure serves as a driving force in macromolecular folding and assembly [79]. SASA provides insights into structural changes, such as the burial of surface area during folding for studying solvation effects and biomolecular behavior [80]. For receptors to maintain stability throughout the MDS, their solvent-accessible surface area should remain consistent, indicating the absence of any significant conformational alterations[. S4 Fig in S1 File. (in the supplementary information) presents the SASA profiles of the protein backbones in the top eight protein-ligand complexes.

The absence of distinct upward or downward trends in the curves indicates that ligand binding does not induce significant conformational changes in the protein's surface structure. SASA for the apo protein was calculated as $219.44 \pm 4.31$ nm$^2$. Upon ligand binding, the SASA for all complexes, including beta-sitosterol ($224.88 \pm 5.43$ nm$^2$), squalene ($223.62 \pm 4.18$ nm$^2$), etretinate ($221.64 \pm 4.49$ nm$^2$), rhoifolin ($220.81 \pm 4.16$ nm$^2$), swertisin ($218.23 \pm 3.79$ nm$^2$), phloridzin ($219.54 \pm 5.11$ nm$^2$), rhapontin ($220.37 \pm 3.98$ nm$^2$), and diosmetin-7-O-beta-D-glucopyranoside ($225.97 \pm 3.22$ nm$^2$), remained within a comparable range to that of the apo protein, with no significant deviations observed. This stability suggests that the tertiary geometry of the protein remained robust, and no folding or unfolding events were triggered upon ligand binding. These findings underscore the structural integrity of the protein during MDS, supporting the notion that the observed interactions are localized without significantly perturbing the overall conformation. Consequently, the consistency in SASA reflects the resilience of the protein's surface and highlights its suitability as a stable receptor for ligand binding without compromising its functional architecture.

### 3.4.5. Radial pair distribution function (RPDF).

RPDF serves as a statistical measure of spatial distribution of the ligand atoms around the protein, where g(r) indicates the probability of finding the ligand's center of mass at a certain

distance (r) from the protein's center of mass [81]. It gives the information about the binding interactions by analyzing the spatial distribution of specific pairs; typically, between the ligand and protein over time. The RPDF plot for the top 8 complexes is illustrated in Fig 6.

Complex 1 (0.96 nm), 2 (0.62 nm), 3 (1.79 nm), and 5 (1.44 nm) showed prominent, sharp peaks at relatively short distances in the RPDF plots, with Complex 2 showing the tallest spike at 0.62 nm, among the complexes. In contrast, Complex 3 displayed the shortest spike at 1.79 nm. This suggests that these ligands are tightly bound to the protein and maintain a consistent distance from it. Complexes 4 (1.08 nm), 6 (0.95 nm), 7 (0.97 nm), and 8 (1.26 nm) exhibited relatively broader peaks. This indicates that these ligands have a wider range of distances from the protein. They might be more flexible in their binding or have greater freedom of movement within the catalytic pocket.

The presence of a single distinct peak for each complex indicated that the ligands consistently occupied a specific spatial region relative to the protein, demonstrating their localized and conserved positioning within the orthosteric binding site throughout the simulation. This observation signifies the stability of their binding poses and highlights their sustained interaction with the protein over time.

**3.4.6. Radius of gyration ($R_g$).** The radius of gyration ($R_g$) is a crucial metric for analyzing protein conformational changes and dynamics during MDS [82]. It quantifies the root mean square deviation of atoms from the center of mass, offering insights into protein compactness and distribution of atoms in the structure [83]. A lower $R_g$ signifies reduced atomic dispersion from the center, indicating a more compact structure [84]. Tracking $R_g$ changes upon complex formation is vital for assessing the stability of protein-ligand interactions. S5 Fig in S1 File. (in the supplementary information) presents the $R_g$ of the proteins within the top eight protein-ligand complex, as well as that for the apo protein.

The radius of gyration ($R_g$) for the proteins in all eight complexes, as well as in the apo protein, ranged from 2.36 to 2.39 nm. Specifically, the $R_g$ for Complex 1, Complex 2, Complex 3, Complex 4, Complex 5, Complex 6, Complex 7, and Complex 8 were 2.38±0.01 nm, 2.38±0.01 nm, 2.37±0.01 nm, 2.36±0.01 nm, 2.36±0.01 nm, 2.37±0.01 nm, 2.37±0.01

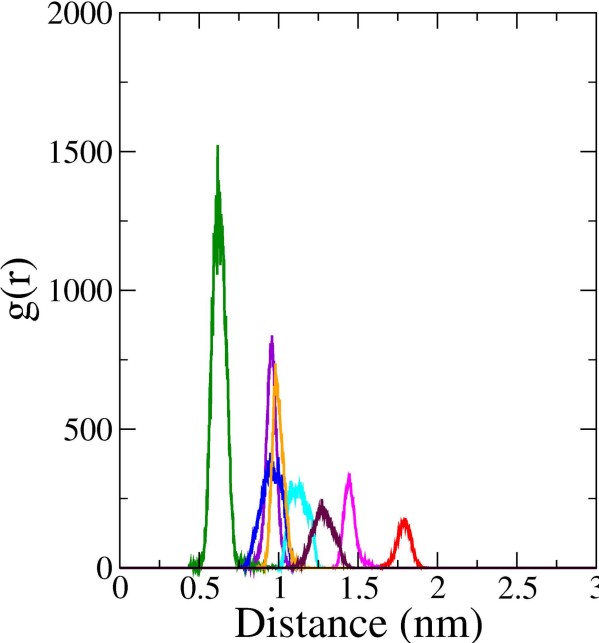

**Fig 6. RPDF plot depicting the center of mass between the top 8 ligands and the protein within these complexes extracted from the MDS trajectories; a distinct sharp peak indicates the ligand's localized positioning; Complex 1 (violet); Complex 2 (green); Complex 3 (red); Complex 4 (cyan); Complex 5 (magenta); Complex 6 (blue); Complex 7 (orange); Complex 8 (maroon).**

nm, and 2.38 ± 0.08 nm, respectively. These values across the complexes showed minimal differences, with Complex 8 exhibiting slightly greater fluctuations in structural compactness. The apo protein showed an $R_g$ of 2.38 ± 0.01 nm, identical to that of Complex 1 and 2, indicating that ligand binding did not significantly alter the overall compactness of the protein structure. The curves for the apo protein (black) and the protein in Complex 7 (orange) displayed similar trends, showing a smooth profile up to around 100 ns. After this point, both curves exhibited a slight downward shift, which remained consistent until the final 100 ns of the simulation. However, the protein in Complex 7 showed less fluctuation after 100 ns compared to the apo protein, indicating a more stable structural conformation in the complex. Similar curves with moderate smoothness were observed for the remaining complexes, showing consistent trends throughout the simulation.

These results suggest that the proteins in all complexes, as well as the apo protein, maintained a similar level of compactness, with only minor variations in $R_g$, reflecting a generally stable and consistent structural conformation across different complexes.

**3.4.7. H-Bond count and H-bond distribution.** H-bonds play a vital role in establishing and maintaining the structural integrity of proteins and the complexes. These bonds continuously form and break as proteins adapt their structure, for example, when shifting from a non-functional to functional states [85] owing to their pronounced directionality, short-range interactions, and abundance in folded proteins. This highlights the importance of quantifying the hydrogen bonds formed between a ligand and its target, as they play a critical role in the drug discovery process by contributing to binding affinity, specificity, and overall molecular stability. From the MDS of the top eight protein-ligand complexes, it was observed that the highest number of hydrogen bonds were formed in Complex 5 with swertisin as indicated by the smooth progression of the ligand RMSD curve in Fig 5, while no hydrogen bonds were detected in Complex 2 with squalene. The strength of the hydrogen bonds formed must also be analyzed to evaluate their contribution to the efficiency and stability of the complexes (more persistent and stronger hydrogen bonds suggest a more stable and efficient complex, enhancing binding and functional activity). S6 Fig in S1 File. (in the supplementary information) represents the hydrogen bond (H-bond) count over the simulation time of 200 ns in various complexes.

For Complex 1, the count ranged from 1 to 2, indicating that at any given time, a maximum of two hydrogen bonds were observed. Most of the time, 1 hydrogen bond was consistently present, suggesting a relatively stable interaction. Occasional spikes to 2 hydrogen bonds were observed, indicating transient formation of additional interactions. Some brief instances of gap in the hydrogen bond count curve, might suggest that the ligand temporarily lost contact or reoriented itself within the binding pocket. For Complex 3, for most of the simulation time, 1 hydrogen bond was consistently observed. Occasional formation of 2 or 3 H-bonds was observed intermittently. The periodic formation of 2–3 hydrogen bonds could signify dynamic binding, where the ligand interacts with multiple residues depending on its orientation or the protein's conformational changes. For Complex 4, the number of H-bonds exhibited the cause of significant fluctuations throughout the simulation. This was expected, as H-bonds are dynamic interactions that constantly form and break in molecular systems. The H-bond count appeared to oscillate between approximately 1 and 6. Similar nature was observed for Complex 5, with H-bond count ranged from 1 to 5. Similarly for complex 6, 7, and 8, the H-bond count ranged from 1–6, 1–6, and 1–7 respectively. S7 Fig in S1 File. (in the supplementary information), represents the H-bond distribution at certain distances between the hydrogen bond donor and acceptor atoms in nanometers for top 8 protein-ligand complexes (H-bond absent in Complex 2).

The range in the plots observed was approximately 0.25 nm to 0.35 nm, which falls within the typical hydrogen bond range (2.5–3.5 Å or 0.25–0.35 nm). The single peak of the distribution curve for the complexes indicates low variability in the donor-acceptor distances.

**3.4.8. Hydrogen bond modulation.** Fig 7 and 8 depicts the variations in the donor-acceptor distance of the hydrogen bond established between the ligand and the protein throughout the simulation. In Complex 5, HIS90 formed a moderately stable hydrogen bond with the oxygen acceptor of the ligand, exhibiting an average donor-acceptor (D-A) distance of 0.30 ± 0.03 nm. This interaction remained relatively stable, maintaining equilibrium throughout the entirety of the production run.

 

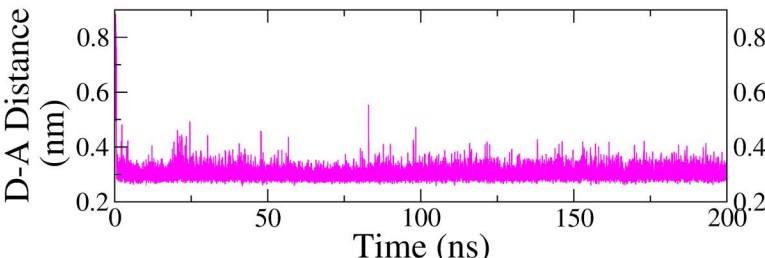

**Fig 7. Changes in D-A atom distances in hydrogen bonds observed during the MDS trajectory for protein-ligand complex with swertisin, involving HIS90 and the oxygen acceptor of the ligand.**

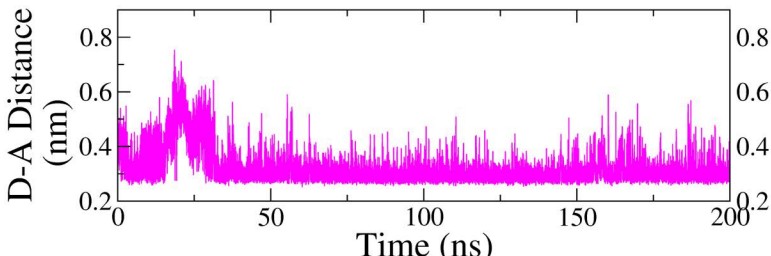

**Fig 8. Changes in D-A atom distances in hydrogen bonds observed during the MDS trajectory for protein-ligand complex with swertisin, involving TYR435 and the oxygen acceptor of the ligand.**

Similarly, in the same complex, the amino acid residue TYR435 formed a moderately stable hydrogen bond, which persisted throughout the production run. The average donor-acceptor (D-A) distance was $0.32 \pm 0.06$ nm, with slight fluctuations at *ca*. 15–30 ns. For the other complexes, the hydrogen bond distance curves exhibited significant fluctuations without a consistent pattern, indicating the absence of stable hydrogen bonds. Consequently, they were excluded from the analysis.

### 3.5. Energetic analysis of protein-ligand adduct formation

The Gibbs free energy of binding change (ΔG) is a fundamental thermodynamic parameter that determines the stability and favorability of protein-ligand complex formation [86]. Protein-ligand binding occurs spontaneously when ΔG is negative, signifying the feasibility of the formation of a stable complex. The magnitude of negative ΔG reflects the strength and stability of the binding, depending solely on the initial and final thermodynamic states, independent of the pathway taken [63]. Table 3 presents the binding free energy changes of the complexes, calculated from the equilibrated segment of the MDS trajectory. Complex 4 displayed the lowest value, $\Delta G_{BFE}$ of $-40.57 \pm 5.18$ kcal/mol, indicating a high level of thermodynamic stability. Similarly, the binding free energy changes of complexes 1, 2, 3, 5, 6, 7, and 8 were $-37.14 \pm 3.91$, $-38.69 \pm 3.61$, $-36.48 \pm 2.58$, $-32.70 \pm 3.21$, $-18.80 \pm 4.45$, $-28.34 \pm 4.27$, and $-28.66 \pm 4.17$ kcal/mol, respectively.

The formation of the complex is opposed by the Poisson-Boltzmann solvation energy, the only positive term. However, the negative contributions from van der Waals interactions (the predominant factor), electrostatic forces, and non-polar interactions surpass the Poisson-Boltzmann energy, ultimately favoring the complex formation. Additionally, S8 Fig in S1 File. (in the supplementary information), provides a detailed frame-by-frame analysis of the binding free energy change ($\Delta G_{BFE}$) across eight distinct complexes throughout the equilibrated phase. The moving average of the binding free energy change consistently remained negative for all complexes indicating sustained spontaneity of the adduct formation. The

**Table 3. Binding free energy changes (kcal/mol) of the top eight different complexes with its components.**

| Complexes | ΔE$_{VDW}$ | ΔE$_{EL}$ | ΔE$_{PB}$ | ΔE$_{NPOLAR}$ | ΔG$_{BFE}$ |
|---|---|---|---|---|---|
| Complex 1 | −57.87 ± 2.29 | −4.54 ± 2.60 | 30.87 ± 2.65 | −5.61 ± 0.08 | −37.14 ± 3.91 |
| Complex 2 | −70.21 ± 2.51 | −1.92 ± 1.31 | 40.36 ± 2.90 | −6.92 ± 0.16 | −38.69 ± 3.61 |
| Complex 3 | −53.01 ± 2.08 | −6.13 ± 2.09 | 28.07 ± 2.31 | −5.41 ± 0.11 | −36.48 ± 2.58 |
| Complex 4 | −71.14 ± 3.53 | −38.17 ± 6.06 | 74.87 ± 4.19 | −6.13 ± 0.10 | −40.57 ± 5.18 |
| Complex 5 | −54.38 ± 2.66 | −25.36 ± 4.39 | 52.25 ± 3.60 | −5.22 ± 0.08 | −32.70 ± 3.21 |
| Complex 6 | −52.90 ± 2.42 | −21.93 ± 5.41 | 61.24 ± 4.58 | −5.22 ± 0.10 | −18.80 ± 4.45 |
| Complex 7 | −58.26 ± 2.77 | −27.07 ± 5.67 | 62.02 ± 5.04 | −5.03 ± 0.08 | −28.34 ± 4.27 |
| Complex 8 | −59.38 ± 3.46 | −21.18 ± 7.42 | 57.21 ± 5.81 | −5.30 ± 0.11 | −28.66 ± 4.17 |

Where, **ΔG$_{BFE}$** = Change in Gibb's binding free energy, ΔE$_{NPOLAR}$ = Non-polar energy change, ΔE$_{PB}$ = Polar contributions in the solvent-solute system, ΔE$_{EL}$ = Electrostatic energy change, ΔE$_{VDW}$ = van der Waals energy change.

data collectively suggest that the adducts are stable and that the hit compounds have the potential to inhibit the activity of the MAO-B receptor. These theoretical insights provide a foundation for subsequent experimental validation within the drug design and discovery pipeline.

### 3.6. Drug-likeness and ADMET assessment

The high failure rate of drug candidates in clinical trials is predominantly linked to inadequate pharmacokinetic properties or unacceptable toxicity profiles [48,87]. Consequently, the concept of drug-likeness has emerged as a critical parameter in early-stage drug discovery, facilitating the prioritization of compounds with favorable bioavailability [87]. An optimal drug candidate must not only exhibit potent target specific activity but also possess well balanced ADMET characteristics at therapeutically relevant concentrations [87]. Four of the five top-scoring ligands; beta-sitosterol, squalene, etretinate, and swertisin complied with Lipinski's Rule of Five, reflecting favorable drug-likeness comparable to the native ligand and reference MAO-B inhibitors. Despite rhoifolin being an exception, the majority exhibited properties consistent with clinically viable compounds. Rhoifolin's violation of Lipinski's criteria is primarily attributed to its elevated molecular weight and excessive hydrogen bond donors and acceptors, which may be mitigated through targeted structural optimization to enhance its drug-likeness [88]. A detailed comparison of physicochemical properties is provided in the supplementary information (S2 Table in S1 File), followed by a summary of ADMET profiles for the top five ligands, the native ligand, and three clinically relevant MAO-B inhibitors (S3 Table in S1 File).

Based on pkCSM's validated ADMET prediction, beta-sitosterol and squalene demonstrated favorable absorption (HIA > 89%), significant blood-brain barrier penetration (logBB > 0.78; threshold > 0.3), and adequate CNS permeability (logPS ≤ −1.70; threshold > −2), supporting their potential as CNS-targeted therapeutics. Both compounds exhibited no cytochrome P450 inhibition, reducing drug-interaction risks and displayed moderate to high metabolic clearance (logCL = 0.62 and 1.79 mL/min/kg, respectively; thresholds: > 1.0 = high, 0.3–1.0 = moderate), suggesting once to twice daily dosing feasibility. Critically, they showed no predicted hepatotoxicity, cardiotoxicity (hERG), mutagenicity (AMES), or skin sensitization. In contrast, rhoifolin and swertisin were found unsuitable for CNS targets due to poor absorption (HIA 24–51%), negligible brain exposure (logBB ≤ −1.56, logPS ≤ −3.90), and P-gp efflux liability. Etretinate, while well-absorbed (HIA 95.94%) and moderately brain-penetrant (logBB = 0.23), displayed hepatotoxicity risk. These predictions are consistent with the known pharmacokinetic and metabolic limitations approved MAO-B inhibitors; for instance, safinamide exhibited limited CNS permeability (logPS = −2.70) while rasagiline and L-deprenyl are documented inhibitors of CYP1A2, underscoring that even clinically utilized compounds are not devoid of ADMET-related constraints. Collectively, squalene and beta-sitosterol represent promising candidates with optimized ADMET properties for further development as

MAO-B inhibitors, offering enhanced metabolic stability and safety over existing therapeutics like rasagiline (skin sensitizer) or deprenyl (hepatotoxic) (S3 Table in S1 File).

Squalene and beta-sitosterol were identified as optimal MAO-B inhibitor candidates, exhibiting superior CNS bioavailability, favorable clearance, and minimal toxicity or drug-interaction risks, compared to the existing drugs; safinamide, deprenyl, and rasagiline as validated by pkCSM-based ADMET profiling. Comparable conclusions have been reported in previous studies [89,90]. Etretinate, rhoifolin, and swertisin demonstrated partially favorable ADMET profiles, highlighting them as other promising drug candidates. However, they exhibit notable pharmacokinetic limitations, including low predicted BBB permeability, limited CNS penetration, and recognition as substrates of P-glycoprotein, which may restrict their effectiveness in CNS-targeted therapies. Despite these challenges, these compounds showed strong binding affinity and stable interactions with MAO-B. This suggests potential utility as peripheral MAO-B modulators, an approach supported by evidence linking elevated platelet-associated MAO-B activity to Alzheimer's disease [91]. To enhance their suitability for CNS applications, rational structural optimization is necessary. Future modifications should aim to improve BBB permeability and CNS bioavailability while maintaining low toxicity and minimizing the risk of drug-drug interactions. Such optimization could enhance the pharmacological potential of these phytochemicals for therapeutic use.

Nevertheless, it is important to acknowledge the inherent limitations of relying solely on computational methods in drug discovery. While these approaches provide valuable preliminary insights into molecular behavior, pharmacokinetics, and potential efficacy, they are subject to potential inaccuracies arising from model assumptions and simplifications. Molecular docking, in particular, has a limited ability to comprehensively represent the conformational flexibility of both receptors and ligands, which can affect the accuracy of predicted binding poses and affinities. MDS, although useful for exploring biomolecular dynamics, are constrained by short simulation timescales, computational demands, and sensitivity to initial conditions, potentially limiting their capacity to capture long-term conformational changes. Additionally, computational techniques face challenges in fully capturing the complexity and dynamic nature of biological systems. Therefore, predictions derived from molecular docking, molecular dynamics simulations, ADMET profiling, and related *in silico* methods remain theoretical and must be corroborated through rigorous experimental validation to confirm their clinical relevance and therapeutic potential.

## 4. Conclusion

Molecular docking of 36 *Oxalis* phytochemicals against MAO-B enzyme, identified five top candidates; beta-sitosterol (−11.92 kcal/mol), squalene (−11.89 kcal/mol), etretinate (−11.46 kcal/mol), rhoifolin (−11.44 kcal/mol), swertisin (−11.13 kcal/mol) with binding affinities surpassing that of the native ligand (−11.12 kcal/mol), with beta-sitosterol exhibiting the highest affinity. Additionally, three compounds; phloridzin, rhapontin, and diosmetin 7-O-beta-D-glucopyranoside exhibited binding affinities between −10.96 and −11.10 kcal/mol, exceeding those of the reference drugs (MAO-B inhibitors; Safinamide, L-deprenyl, rasagiline). Hydrophobic interactions played a dominant role in stabilizing the protein–ligand complexes, with additional stabilization provided by hydrogen bonding and π-stacking. This might contribute to the functional inhibition of the enzyme, acting through an antagonistic mechanism. Molecular dynamics simulations confirmed the structural stability of the protein-ligand complexes, as indicated by small RMSD and RMSF. SASA, and $R_g$ analyses suggested that ligand binding preserved the protein's structural integrity. Additionally, ligand RMSD and simulation snapshots showed that the ligand remained consistently positioned within the binding site. This stability was further supported by hydrogen bond network, compact protein conformation reflected in $R_g$, and distinct single peaks in the RPDF plots, reinforcing sustained active-site interactions. Binding free energy change calculations confirmed the thermodynamic feasibility of the protein-ligand adduct formation (−40.57 ± 5.18 to −28.34 ± 4.27 kcal/mol) with rhoifolin exhibiting the best change (ΔG = −40.57 ± 5.18 kcal/mol). ADMET prediction highlighted squalene and beta-sitosterol as promising CNS-active MAO-B inhibitors with favorable pharmacokinetics and safety profiles, while etretinate, rhoifolin, and swertisin may serve as peripheral modulators, requiring structural optimization to improve CNS accessibility and therapeutic

potential. These findings collectively indicate that *Oxalis* phytochemicals exhibit promising MAO-B inhibitory potential, with beta-sitosterol, rhoifolin, and swertisin being possibly the strong drug candidates for the therapeutic applications in age related neurodegenerative diseases such as Alzheimer's and Parkinson's. Subsequent experimental studies (both *in vitro* and *in vivo*), are required to validate their efficacy, pharmacokinetics, and safety, in order to advance their potential as plant-derived neuroprotective agents.

## Supporting information

**S1 File. S1 Fig. Binding modes and interactions of ligands with MAO-B: Ribbon model representation (left) showing the orthosteric pocket of MAO-B (PDB ID: 4A79) and 3D interaction profiles (right) illustrating the interacting amino acid residues of the enzyme with the top 8 ligands, along with the native ligand. S2 Fig. RMSD of protein backbone complexed with top 8 ligands relative to protein backbone along with apo protein (black); Complex 1; violet=beta-sitosterol, Complex 2; green=squalene, Complex 3; red=etretinate, Complex 4; cyan=rhoifolin, Complex 5; magenta=swertisin, Complex 6; blue=phloridzin, Complex 7; orange=rhapontin, Complex 8; maroon=diosmetin-7-O-beta-D-glucopyranoside). S3 Fig. RMSF curves of alpha-carbon atoms of protein backbone in top 8 protein-ligand complexes relative to the protein backbone along with that of apo protein (black); Complex 1; violet=beta-sitosterol, Complex 2; green=squalene, Complex 3; red=etretinate, Complex 4; cyan=rhoifolin, Complex 5; magenta=swertisin, Complex 6; blue=phloridzin, Complex 7; orange=rhapontin, Complex 8; maroon=diosmetin-7-O-beta-D-glucopyranoside. S4 Fig. Variation of SASA of proteins in top 8 protein ligand complexes compared to that of the apo protein (black). (Complex 1; violet=beta-sitosterol, Complex 2; green=squalene, Complex 3; red=etretinate, Complex 4; cyan=rhoifolin, Complex 5; magenta=swertisin, Complex 6; blue=phloridzin, Complex 7; orange=rhapontin, Complex 8; maroon=diosmetin-7-O-beta-D-glucopyranoside). S5 Fig. Variation of the radius of gyration of the protein in top 8 protein-ligand complexes and the apo protein (black) obtained from the MDS trajectories; Complex 1 (violet); Complex 2 (green); Complex 3 (red); Complex4 (cyan); Complex 5 (magenta); Complex 6 (blue); Complex 7 (orange); Complex 8 (maroon). S6 Fig. Variation in hydrogen bond count in the top 8 protein ligand complexes (absent in Complex 2) throughout the MDS; (1) Complex 1 (violet); (3) Complex 3 (red); (4) Complex 4 (cyan); (5) Complex 5 (magenta); (6) Complex 6 (blue); (7) Complex 7 (orange); (8) Complex8 (maroon). S7 Fig. The frequency of donor-acceptor distance in hydrogen bonds within the top 8 protein ligand complexes (no H-Bond in Complex 2) during MDS; (1) Complex 1 (violet); (3) Complex 3 (red); (4) Complex 4 (cyan); (5) Complex 5 (magenta); (6) Complex 6 (blue); (7) Complex 7 (orange); (8) Complex 8 (maroon). S8 Fig. Variational curves of binding free energy change for the top 8 protein-ligand adducts across the equilibrated segment of MDS trajectories; persistently negative moving averages, confirms the steady spontaneity throughout the production run. S1 Table. Binding affinity of phytocompounds present in *Oxalis corniculata* and *Oxalis latifolia.* S2 Table. Physiochemical properties of the top ligands along with that of native ligand and reference drugs**. **S3 Table. ADMET profiling of top five ligands along with that of native ligand and reference drugs considered.**
(DOCX)

## Author contributions

**Conceptualization:** Bishnu P. Marasini, Jhashanath Adhikari Subin.

**Data curation:** Ram Lal (Swagat) Shrestha, Shiva M.C..

**Formal analysis:** Ram Lal (Swagat) Shrestha, Shiva M.C., Ashika Tamang, Manila Poudel, Nirmal Parajuli, Bishnu P. Marasini.

**Investigation:** Ram Lal (Swagat) Shrestha, Shiva M.C..

**Methodology:** Ram Lal (Swagat) Shrestha, Shiva M.C..

**Resources:** Aakar Shrestha, Timila Shrestha, Samjhana Bharati, Binita Maharjan.

**Software:** Aakar Shrestha, Timila Shrestha, Samjhana Bharati, Binita Maharjan.

**Supervision:** Bishnu P. Marasini, Jhashanath Adhikari Subin.

**Validation:** Jhashanath Adhikari Subin.

**Visualization:** Ashika Tamang, Manila Poudel, Nirmal Parajuli.

**Writing – original draft:** Ram Lal (Swagat) Shrestha, Shiva M.C..

**Writing – review & editing:** Jhashanath Adhikari Subin.

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
