## [Decision Letter · Decision Letter 0]

Dear Dr. Adhikari Subin,

Thank you for submitting your manuscript to PLOS ONE. After careful consideration, we feel that it has merit but does not fully meet PLOS ONE’s publication criteria as it currently stands. Therefore, we invite you to submit a revised version of the manuscript that addresses the points raised during the review process.

**ACADEMIC EDITOR:**

We look forward to receiving your revised manuscript.

Kind regards,

Yusuf Oloruntoyin Ayipo, Ph.D

Academic Editor

PLOS ONE

Journal Requirements:

3. We note you have included a table to which you do not refer in the text of your manuscript. Please ensure that you refer to Table 4 in your text; if accepted, production will need this reference to link the reader to the Table.

**Additional Editor Comments:**

The study design reflects scientific relevance and the manuscript is well -composed. However, some concerns need to be addressed before the manuscript can be considered for publication in this journal. For instance, in the discussion lines 201=207, the authors need to reference the facts stated regarding the observed interactions between the selected ligands and amino acid residues at the active site of the enzyme. Other concerns raised by the respective reviewers also deserve a substantial attention.

Reviewers' comments:

Reviewer's Responses to Questions

**Comments to the Author**

1. Is the manuscript technically sound, and do the data support the conclusions?

Reviewer #1: Partly

Reviewer #2: Yes

Reviewer #3: Yes

2. Has the statistical analysis been performed appropriately and rigorously?

Reviewer #1: Yes

Reviewer #2: Yes

Reviewer #3: N/A

3. Have the authors made all data underlying the findings in their manuscript fully available?

Reviewer #1: No

Reviewer #2: Yes

Reviewer #3: No

4. Is the manuscript presented in an intelligible fashion and written in standard English?

Reviewer #1: Yes

Reviewer #2: Yes

Reviewer #3: Yes

Reviewer #1: I am unable to access the figures, Please resubmit figures.

Secondly Molecular docking has following limitations :

1. Scoring Function Limitations

2. Rigid Docking

3. Lack of Entropy and Solvent Effects

4. Challenges with Protein Flexibility

5.Protein Dynamics

6. Ligand Conformational Changes

You didnt not mention the limitations of this approch.Please discuss that

Reviewer #2: Congratulations on your manuscript titled; “Evaluating the Inhibitory Efficacy of Oxalis Phytocompounds on Monoamine Oxidase B: An Integrated Approach Targeting Age Related Neurodegenerative Diseases through Molecular Docking and Dynamics Simulations.”

To improve its suitability for publication, I will recommend working on the references & Figure citations and other points outlined below.

Introduction

1. Wilson et al., 2023 should be cited as number 1 reference and update others appropriately.

Molecular interactions in different adducts

2. Cite the complexes structure in the SI: “Notably, hydrophobic interactions were found as the predominant type of interactions in Complex 1, Complex 2, Complex 3, and Complex 5 (see supporting information)”

Geometrical Analysis of Adducts Conformation

3. Provide an appropriate citation and reference for; “The adduct is considered more stable if the ligand stays bound to the orthosteric site for a longer duration while maintaining its orientation and location relative to the protein backbone”

4. Specify which complex contains Beta-sitosterol; “In complex X, Beta-sitosterol exhibited only minor orientation changes at 50, 100, 150, and 200 ns compared to its position at 1 ns in the adduct”

Root mean square deviation (RMSD)

5. Provide an appropriate citation and reference for; “. A higher RMSD suggests greater deviation from the reference conformation over time, while a stable RMSD suggests system equilibration, and persistent fluctuations may imply ongoing structural adjustments”

H-Bond Count and H-bond distribution

6. Include the image for complex 2 in Fig S6, especially as it shows no Hydrogen bonds.

Energetic analysis of protein-ligand adduct formation

7. Remove word editor box around the text; “Table 2 presents the binding free energy changes of the complexes, calculated from the equilibrated segment of the MDS trajectory.”

8. Remove the text if not applicable“ ∆GGas= Gas phase energy change, ∆GSol = Electrostatic solvation energy change”

9. Fig S8 image is missing in the supporting information.

10. Remove word editor box around the text; “Additionally, Fig S8 in the supplementary information, provides a detailed frame-by-frame analysis of the binding free energy change (∆GBFE) across eight distinct complexes throughout the equilibrated phase. It revealed that the moving average consistently stayed within the negative range”

11. Rewrite the entire paragraph for clarity: “It revealed that the moving average consistently stayed within the negative range for all complexes indicating sustained spontaneity of the adduct formation reaction. All the data pointed towards stable nature of the adducts and the hit candidates are possibly capable of inhibiting the functioning of the receptor MAO-B. The inferences derived from theoretical procedure forms the basis for experimental validation in the drug design and discovery procedural pipeline. “

Reviewer #3: Reviewer Note:

Aim of the Paper:

The primary goal of the study was to perform molecular docking and dynamic simulations to evaluate the inhibitory potential of phytochemicals from Oxalis species against monoamine oxidase B (MAO-B), in support of treating neurodegenerative diseases such as Alzheimer’s disease.

Aim Achieved:

Yes, the paper identified five compounds; beta-sitosterol, squalene, etretinate, rhoifolin, and swertisin and showed that they have potential to bind MAO-B

Abstract:

The abstract is comprehensive and well-detailed.

Paper Claims and Significance:

The paper claims that the identified compounds exhibit better binding affinity than known reference drugs. This is significant, as neurodegenerative diseases remain a critical public health challenge. However, the overall impact of the study is limited due to its complete reliance on in-silico methods.

Introduction:

The introduction provides an excellent overview of the research topic and offers a compelling justification for selecting phytochemicals. It also discusses current therapies.

A more thorough review of previous MAO-B inhibition studies involving similar phytochemicals would strengthen the rationale for selecting compounds from Oxalis.

Results:

The docking scores, simulations, and binding free energy analyses are well-executed and clearly presented.

The data presented in the manuscript support the central claim that several phytocompounds bind MAO-B stably and with high affinity.

Limitations:

No in vitro validation of the identified phytocompounds, which limits the study’s impact.

No ADMET modeling was provided.

Only one crystal structure (PDB ID: 4A79) was used; cross-validation with other MAO-B structures would strengthen the findings.

Recommendation:

The study is well-conceived and methodologically sound for an in-silico paper. However, please consider the following suggestions:

Include ADMET modeling.

Cross-validate the use of PDB ID: 4A79, or provide a justification for selecting only one crystal structure.

Discuss the limitations of computational-only approaches.

Compare the results with known MAO-B inhibitors more critically.

Provide docking and simulation input files as supplementary materials.

**Do you want your identity to be public for this peer review?** For information about this choice, including consent withdrawal, please see our Privacy Policy

Reviewer #1: **Yes: ** Sammuel Shahzad

Reviewer #2: No

Reviewer #3: No

---

## [Author Response · Author response to Decision Letter 1]

13 Jun 2025

Dear Editor,

We would like to thank you and the reviewers for the constructive and valuable feedback. We

have attempted to incorporate all the suggestions to the best of knowledge and skills. It has

helped to fix the errors and fulfill the insufficiency to uplift the scientific content of the

manuscript.

Please let us know in case the responses are unclear or insufficient. The point-by-point responses are mentioned below.

Sincerely,

Authors

Evaluating the inhibitory efficacy of Oxalis phytocompounds on monoamine oxidase B: An integrated approach targeting age related neurodegenerative diseases through molecular docking and dynamics simulations

ACADEMIC EDITOR:

The study design reflects scientific relevance and the manuscript is well -composed. However, some concerns need to be addressed before the manuscript can be considered for publication in this journal. For instance, in the discussion lines 201=207, the authors need to reference the facts stated regarding the observed interactions between the selected ligands and amino acid residues at the active site of the enzyme. Other concerns raised by the respective reviewers also deserve a substantial attention.

Response: Regarding lines 201–207, we have revised the discussion section to include appropriate references that support the observed interactions between the selected ligands and key active site residues of MAO-B. In addition, data from the BioLiP server have been incorporated to further substantiate these findings. We have also carefully addressed all other concerns raised by the reviewers in the revised manuscript.

We note you have included a table to which you do not refer in the text of your manuscript. Please ensure that you refer to Table 4 in your text; if accepted, production will need this reference to link the reader to the Table.

Response: Table 4 is appropriately cited and discussed in the main text of the revised manuscript to guide the reader effectively.

Reviewer 1:

I am unable to access the figures, Please resubmit figures.

Response: We sincerely apologize for the inconvenience caused. All the figures have been carefully reviewed and resubmitted in high-resolution format and appropriate file types to ensure full accessibility. We have verified that the revised submission meets the journal’s technical specifications for figure formatting and clarity. Please let us know if any further adjustments are required.

Secondly Molecular docking has following limitations:

1. Scoring Function Limitations

2. Rigid Docking

3. Lack of Entropy and Solvent Effects

4. Challenges with Protein Flexibility

5. Protein Dynamics

6. Ligand Conformational Changes

You didnt not mention the limitations of this approach. Please discuss that.

Response: We have revised the manuscript to include a detailed discussion of the inherent limitations of molecular docking, specifically addressing the issues raised, including scoring function accuracy, treatment of protein and ligand flexibility, entropy and solvation effects, and conformational dynamics. This discussion has been incorporated in the revised manuscript (in the Computational Method, Molecular Docking Calculations section), where we also justify the selection of the DockThor docking platform. DockThor's soft docking protocol partially addresses these limitations by allowing both protein and ligand flexibility within a hydrated environment, thus improving the biological relevance of the docking results. However, we also acknowledge that, like most docking tools, DockThor's scoring function remains a limitation, and this is clearly stated in the revised text.

Reviewer 2: Congratulations on your manuscript titled; “Evaluating the Inhibitory Efficacy of Oxalis Phytocompounds on Monoamine Oxidase B: An Integrated Approach Targeting Age Related Neurodegenerative Diseases through Molecular Docking and Dynamics Simulations.”

To improve its suitability for publication, I will recommend working on the references & Figure citations and other points outlined below.

Introduction

1. Wilson et al., 2023 should be cited as number 1 reference and update others appropriately.

Response: We have updated the reference list so that Wilson et al., 2023 now appears as Reference [1]. All in-text citations and the bibliography have been carefully renumbered to ensure consistency throughout the manuscript.

Molecular interactions in different adducts

2. Cite the complexes structure in the SI: “Notably, hydrophobic interactions were found as the predominant type of interactions in Complex 1, Complex 2, Complex 3, and Complex 5 (see supporting information)”

Response: The manuscript has been revised to explicitly cite the relevant figure in the Supporting Information. The sentence has been updated to:

Notably, hydrophobic interactions were found as the predominant type of interactions in Complex 1, Complex 2, Complex 3, and Complex 5 (as illustrated in the protein-ligand interaction profiling presented in Fig. S1 of the Supporting information).

Geometrical Analysis of Adducts Conformation

3. Provide an appropriate citation and reference for; “The adduct is considered more stable if the ligand stays bound to the orthosteric site for a longer duration while maintaining its orientation and location relative to the protein backbone”

Response: “The adduct is considered more stable if the ligand stays bound to the orthosteric site for a longer duration while maintaining its orientation and location relative to the protein backbone” has now been appropriately cited in the revised manuscript. The reference has been added to support the statement and is included in the reference list.

4. Specify which complex contains Beta-sitosterol; “In complex X, Beta-sitosterol exhibited only minor orientation changes at 50, 100, 150, and 200 ns compared to its position at 1 ns in the adduct”

Response: We have revised the manuscript to specify that Beta-sitosterol is present in Complex 1. The text has been clarified as follows.

“In first complex, beta-sitosterol exhibited only minor orientation changes at 50, 100, 150, and 200 ns compared to its position at 1 ns in the adduct.”

Root mean square deviation (RMSD)

5. Provide an appropriate citation and reference for; “A higher RMSD suggests greater deviation from the reference conformation over time, while a stable RMSD suggests system equilibration, and persistent fluctuations may imply ongoing structural adjustments”

Response: The statement “A higher RMSD suggests greater deviation from the reference conformation over time, while a stable RMSD suggests system equilibration, and persistent fluctuations may imply ongoing structural adjustments” has now been supported with appropriate references and citations.

H-Bond Count and H-bond distribution

6. Include the image for complex 2 in Fig S6, especially as it shows no Hydrogen bonds.

Response: We have included the image for Complex 2 in Figure S6 of the Supporting Information, highlighting that it exhibits no hydrogen bonds as noted.

Energetic analysis of protein-ligand adduct formation

7. Remove word editor box around the text; “Table 2 presents the binding free energy changes of the complexes, calculated from the equilibrated segment of the MDS trajectory.”

Response: The editor box around the mentioned text has been removed, and the formatting has been corrected accordingly in the revised manuscript.

8. Remove the text if not applicable“ ∆GGas= Gas phase energy change, ∆GSol = Electrostatic solvation energy change”

Response: The text “∆GGas= Gas phase energy change, ∆GSol = Electrostatic solvation energy change” has been removed.

9. Fig S8 image is missing in the supporting information.

Response: We would like to clarify that Figure S8 was included in the Supporting Information at the time of submission. It may have been inadvertently overlooked, but we have double checked to ensure that it is clearly presented in the revised file.

10. Remove word editor box around the text; “Additionally, Fig S8 in the supplementary information, provides a detailed frame-by-frame analysis of the binding free energy change (∆GBFE) across eight distinct complexes throughout the equilibrated phase. It revealed that the moving average consistently stayed within the negative range”

Response: The editor box surrounding the mentioned text has been removed from the revised manuscript.

11. Rewrite the entire paragraph for clarity: “It revealed that the moving average consistently stayed within the negative range for all complexes indicating sustained spontaneity of the adduct formation reaction. All the data pointed towards stable nature of the adducts and the hit candidates are possibly capable of inhibiting the functioning of the receptor MAO-B. The inferences derived from theoretical procedure forms the basis for experimental validation in the drug design and discovery procedural pipeline. “

Response: The entire paragraph has been rewritten for clarity.

Reviewer 3

Aim of the Paper:

The primary goal of the study was to perform molecular docking and dynamic simulations to evaluate the inhibitory potential of phytochemicals from Oxalis species against monoamine oxidase B (MAO-B), in support of treating neurodegenerative diseases such as Alzheimer’s disease.

Aim Achieved:

Yes, the paper identified five compounds; beta-sitosterol, squalene, etretinate, rhoifolin, and swertisin and showed that they have potential to bind MAO-B

Abstract:

The abstract is comprehensive and well-detailed.

Paper Claims and Significance:

The paper claims that the identified compounds exhibit better binding affinity than known reference drugs. This is significant, as neurodegenerative diseases remain a critical public health challenge. However, the overall impact of the study is limited due to its complete reliance on in-silico methods.

Introduction:

The introduction provides an excellent overview of the research topic and offers a compelling justification for selecting phytochemicals. It also discusses current therapies.

A more thorough review of previous MAO-B inhibition studies involving similar phytochemicals would strengthen the rationale for selecting compounds from Oxalis.

Response: We have revised the Introduction section to include a more detailed discussion of previous MAO-B inhibition studies involving flavonoids and polyphenols structurally similar to those found in Oxalis. This addition strengthens the scientific rationale for the selection of Oxalis-derived compounds in our study. Relevant references have also been cited to support this context.

Results:

The docking scores, simulations, and binding free energy analyses are well-executed and clearly presented.

The data presented in the manuscript support the central claim that several phytocompounds bind MAO-B stably and with high affinity.

Limitations:

No in vitro validation of the identified phytocompounds, which limits the study’s impact.

No ADMET modeling was provided.

Only one crystal structure (PDB ID: 4A79) was used; cross-validation with other MAO-B structures would strengthen the findings.

Recommendation:

The study is well-conceived and methodologically sound for an in-silico paper. However, please consider the following suggestions:

Include ADMET modeling.

Response: We have incorporated ADMET (Absorption, Distribution, Metabolism, Excretion, and Toxicity) analysis into the revised manuscript to enhance the pharmacokinetic and safety profiling of the selected compounds. The results of the ADMET prediction have been added to the appropriate section (Results and Discussion), and discussed accordingly to provide a more comprehensive evaluation of the drug-likeness and potential development of the ligands.

Cross-validate the use of PDB ID: 4A79, or provide a justification for selecting only one crystal structure.

Response: We appreciate the reviewer’s suggestion regarding the structural validation of the chosen target. In the present study, PDB ID: 4A79 was selected as the representative crystal structure based on a redocking validation approach. The native ligand was redocked into the active site of 4A79, and the resulting root-mean-square deviation (RMSD) between the redocked pose and the original co-crystallized conformation was found to be minimal, indicating excellent structural fidelity and docking reproducibility. This low RMSD value confirms the suitability of 4A79 for accurate molecular docking studies. Therefore, 4A79 was chosen as the optimal structure for this work, providing a reliable and validated framework for subsequent binding analyses. However, another MAO-B protein was not employed.

Discuss the limitations of computational-only approaches.

Response: We have carefully considered this point and have addressed it in the revised manuscript. Specifically, we have included a dedicated discussion section (at the end of the Result and Discussion section), highlighting the inherent limitations of relying solely on computational methods, such as potential inaccuracies due to model assumptions, the need for experimental validation, and the challenges in capturing complex real-world phenomena fully through simulations alone. We believe these additions clarify the scope and context of our approach and improve the overall rigor of the study.

Compare the results with known MAO-B inhibitors more critically.

Response: In the revised manuscript, we have conducted a thorough comparative analysis between the top five screened ligands and three known MAO-B inhibitors; L-deprenyl, rasagiline, and safinamide. This comparison includes docking scores, ADMET profiling, and detailed protein-ligand interaction analyses.

Most of the top ligands exhibited higher binding affinities than the reference drugs, indicating potentially stronger inhibitory interactions. Importantly, the interaction patterns of the screened ligands were largely consistent with those of the known inhibitors, involving binding to the same critical amino acid residues within the MAO-B active site. Additionally, several ligands formed additional contacts, suggesting more extensive engagement with the enzyme. The ADMET results further supported the drug-likeness of the top ligands, with two compounds demonstrating particularly favorable pharmacokinetic and safety profiles relevant to age-related neurodegenerative conditions. These findings collectively highlight the promising potential of the screened compounds as competitive and possibly superior MAO-B inhibitors.

Provide docking and simulation input files as supplementary materials.

Response: We have included the relevant input parameter files used for both molecular docking and molecular dynamics simulations (MDS) as supplementary materials in the revised manuscript. Specifically, this includes the docking configuration files and the .mdp files used to define the simulation parameters for MDS.

---

## [Decision Letter · Decision Letter 1]

Dear Dr. Adhikari Subin,

We look forward to receiving your revised manuscript.

Kind regards,

Yusuf Oloruntoyin Ayipo, Ph.D

Academic Editor

PLOS ONE

Journal Requirements:

**Additional Editor Comments:**

Kudos to the authors for responding positively to the initial queries. No doubt, the quality of the submission has improved significantly. However, some concerns have been raised affecting some sections of the manuscript. I hereby recommend another round of revision to address the current concerns and reserve my final decision until they are resolved.

Reviewers' comments:

Reviewer's Responses to Questions

**Comments to the Author**

Reviewer #1: (No Response)

Reviewer #2: (No Response)

Reviewer #3: All comments have been addressed

2. Is the manuscript technically sound, and do the data support the conclusions?

Reviewer #1: Partly

Reviewer #2: Yes

Reviewer #3: Yes

3. Has the statistical analysis been performed appropriately and rigorously?

Reviewer #1: I Don't Know

Reviewer #2: Yes

Reviewer #3: Yes

4. Have the authors made all data underlying the findings in their manuscript fully available?

Reviewer #1: No

Reviewer #2: Yes

Reviewer #3: Yes

5. Is the manuscript presented in an intelligible fashion and written in standard English?

Reviewer #1: Yes

Reviewer #2: Yes

Reviewer #3: Yes

Reviewer #1: In this research, objective was to tackle the increasing incidence of neurodegenerative disorders such as Alzheimer’s and Parkinson’s, which are becoming more prevalent in older populations. Acknowledging the essential function of monoamine oxidase B (MAO-B) in the metabolism of dopamine and the management of oxidative stress, Authors examined plant-based compounds from Oxalis species as safer and more affordable alternatives to traditional MAO-B inhibitors. While current synthetic medications may be effective, they frequently come with safety issues and high costs. Utilizing advanced computational methods, we assessed the therapeutic potential of these natural compounds. Through this research, authors highlight the importance of combining traditional medicinal practices with contemporary drug discovery technologies to facilitate the development of natural, neuroprotective approaches for neurological disorders associated with aging.

Please address following Concerns:

1. I am unable to access the Figure 1. Please provide figure 1 as mentioned in Line 93.similarly i cant see the figure 2,3,4and 5

2. Please discuss the limitations of the computational methodoligies in drug discovery. For example please list limitations of Molecular docking and MDS

3.How these computational methodologies are better than the bench work and how each predicted target can be biologically tested.

4. Please discuss in detail the validations of the predicted targets

5. Explain the docking score

Reviewer #2: Most of my comments have been fully addressed, with the exception of:

H-Bond Count and H-bond distribution

6. Include the image for complex 2 in Fig S6, especially as it shows no Hydrogen bonds. (This update still appears to be missing from the revised Supporting Information).

Reviewer #3: (No Response)

**Do you want your identity to be public for this peer review?** For information about this choice, including consent withdrawal, please see our Privacy Policy

Reviewer #1: **Yes: ** Sammuel Shahzad

Reviewer #2: No

Reviewer #3: **Yes: ** Opeoluwa Iwaloye

---

## [Author Response · Author response to Decision Letter 2]

29 Jun 2025

Reviewer 1:

In this research, objective was to tackle the increasing incidence of neurodegenerative disorders such as Alzheimer’s and Parkinson’s, which are becoming more prevalent in older populations. Acknowledging the essential function of monoamine oxidase B (MAO-B) in the metabolism of dopamine and the management of oxidative stress, authors examined plant-based compounds from Oxalis species as safer and more affordable alternatives to traditional MAO-B inhibitors. While current synthetic medications may be effective, they frequently come with safety issues and high costs. Utilizing advanced computational methods, we assessed the therapeutic potential of these natural compounds. Through this research, authors highlight the importance of combining traditional medicinal practices with contemporary drug discovery technologies to facilitate the development of natural, neuroprotective approaches for neurological disorders associated with aging.

Please address following concerns:

1. I am unable to access the Figure 1. Please provide figure 1 as mentioned in Line 93. Similarly, I can’t see the figure 2, 3, 4 and 5.

Response: We apologize for the inconvenience caused by the missing figures. All figures (Fig 1–5) were included in the revised submission as per the PLOS ONE formatting guidelines. It is possible that the figures generated using the PACE server, as recommended by PLOS ONE, may have appeared unreadable or failed to render properly due to a technical issue during the file upload process or in the reviewer’s display. To address this, we have thoroughly reviewed the submission system and re-uploaded all figures, ensuring they are correctly labeled, properly formatted, and fully accessible. For clarity and to facilitate the review process, we have also included all five figures in TIF format directly below in this response letter.

Fig 1. Monoamine Oxidase-B (MAO-B) enzyme structure in its holo form depicted in ribbon representation, with native ligand P1B, Pioglitazone, shown in bond line model at the orthosteric site.

Fig 2. Superimposition of docked ligand (green) derived from the molecular docking calculations, with the native ligand (cyan) present in the crystal structure (heavy atom RMSD =0.768 Å).

Fig 3. Chemical structures of top 8 phytochemicals from Oxalis species identified based on molecular docking scores against MAO-B enzyme

Fig 4. RMSD of top 8 ligands relative to protein backbone obtained from MDS trajectories of different complexes (violet= beta-sitosterol, green= squalene, red= etretinate, cyan= rhoifolin, magenta= swertisin, blue= phloridzin, orange= rhapontin, maroon= diosmetin-7-O-beta-D-glucopyranoside).

Fig 5. RPDF plot depicting the center of mass between the top 8 ligands and the protein within these complexes extracted from the MDS trajectories; a distinct sharp peak indicates the ligand’s localized positioning; Complex 1 (violet); Complex 2 (green); Complex 3 (red); Complex 4 (cyan); Complex 5 (magenta); Complex 6 (blue); Complex 7 (orange); Complex 8 (maroon).

(a)

(b)

Fig 6: Changes in D-A atom distances in hydrogen bonds observed during the MDS trajectory for protein-ligand complex with swertisin, involving a) HIS90; b) TYR435 and the oxygen acceptor of the ligand, swertisin.

Please let us know if any problems persist, and we will be happy to assist further.

2. Please discuss the limitations of the computational methodologies in drug discovery. For example please list limitations of Molecular docking and MDS.

Response: The specific limitations of molecular docking have been thoroughly addressed in the Molecular docking calculation subsection of the Computational method section. Additionally, a broader discussion addressing the limitations of molecular docking, molecular dynamics simulations (MDS), and computational methodologies in general has been incorporated into the concluding paragraph of the Results and Discussion section, immediately preceding the Conclusion.

3. How these computational methodologies are better than the bench work and how each predicted target can be biologically tested.

Response: The advantages of computational methodologies over traditional bench work, along with the approaches for biological validation of predicted targets, have now been addressed in the revised manuscript. This discussion can be found in lines 123 to 133 of the Introduction section.

4. Please discuss in detail the validations of the predicted targets.

Response: The biological relevance of the predicted target was assessed through an extensive review of the literature, which confirmed its well-established involvement in the pathogenesis of neurodegenerative disorders. Although direct experimental validation was not performed in this study, the selection of monoamine oxidase B (MAOB) as the key target is strongly supported by previous research demonstrating its critical role in Alzheimer’s and Parkinson’s diseases, as detailed in the Introduction and Computational method (Target Selection) sections. This provides an indirect yet robust validation of our computational prediction. Nonetheless, we acknowledge that further experimental studies are required to empirically validate these findings and strengthen the conclusions and we have accordingly recommended such investigations.

5. Explain the docking score

Response: The explanation of the docking score has now been included in the Molecular Docking Calculations subsection of the Computational Method section (lines 191–196).

Reviewer 2:

Most of my comments have been fully addressed, with the exception of:

H-Bond Count and H-bond distribution

6. Include the image for complex 2 in Fig S6, especially as it shows no Hydrogen bonds. (This update still appears to be missing from the revised Supporting Information).

Response: The hydrogen bond count plot for Complex 2 has now been included in Figure S6 of the revised Supporting Information. As correctly noted, Complex 2 does not exhibit any hydrogen bonding throughout the simulation, which is reflected by the flat line in the hydrogen bond count plot (2) (Fig S6.). Consequently, a hydrogen bond distribution curve has not been presented for this complex, as no hydrogen bonds were detected during the trajectory.

Reviewer 3:

(No Response)

---

## [Decision Letter · Decision Letter 2]

Evaluating the inhibitory efficacy of Oxalis phytocompounds on monoamine oxidase B: An integrated approach targeting age related neurodegenerative diseases through molecular docking and dynamics simulations

PONE-D-25-18849R2

Dear Dr. Adhikari Subin,

We’re pleased to inform you that your manuscript has been judged scientifically suitable for publication and will be formally accepted for publication once it meets all outstanding technical requirements.

Kind regards,

Yusuf Oloruntoyin Ayipo, Ph.D

Academic Editor

PLOS ONE

Additional Editor Comments (optional):

The study is timely and well-designed. Again, the submission meets the level of scientific rigour required for publication in this title and all the concerns raised by the respective reviewers have been addressed satisfactorily. I hereby recommend the manuscript for publication in the current version.

Reviewers' comments:

Reviewer's Responses to Questions

**Comments to the Author**

Reviewer #1: All comments have been addressed

Reviewer #2: All comments have been addressed

2. Is the manuscript technically sound, and do the data support the conclusions?

Reviewer #1: Yes

Reviewer #2: Yes

3. Has the statistical analysis been performed appropriately and rigorously?

Reviewer #1: Yes

Reviewer #2: Yes

4. Have the authors made all data underlying the findings in their manuscript fully available?

Reviewer #1: Yes

Reviewer #2: Yes

5. Is the manuscript presented in an intelligible fashion and written in standard English?

Reviewer #1: Yes

Reviewer #2: Yes

Reviewer #1: (No Response)

Reviewer #2: Congratulations on your manuscript, Evaluating the inhibitory efficacy of Oxalis phytocompounds on monoamine oxidase B: An integrated approach targeting age related neurodegenerative diseases through molecular docking and dynamics simulations. With the suggested revisions addressed, the study is well-positioned to make a meaningful contribution. I recommend it for publication—well done!

**Do you want your identity to be public for this peer review?** For information about this choice, including consent withdrawal, please see our Privacy Policy

Reviewer #1: No

Reviewer #2: No

---

## [Editor Report · Acceptance letter]

PONE-D-25-18849R2

PLOS ONE

Dear Dr. Adhikari Subin,

I'm pleased to inform you that your manuscript has been deemed suitable for publication in PLOS ONE. Congratulations! Your manuscript is now being handed over to our production team.

Kind regards,

on behalf of

Dr. Yusuf Oloruntoyin Ayipo

Academic Editor

PLOS ONE